# Feature-directed Active Learning for Learning User Preferences

**Sriram Gopalakrishnan, Utkarsh Soni, Subbarao Kambhampati**
Arizona State University

## Abstract

Learning preferences of users over plan traces can be a challenging task given a large number of features and limited queries that we can ask a single user. Additionally, the preference function itself can be quite convoluted and non-linear. Our approach uses feature-directed active learning to gather the necessary information about plan trace preferences. This data is used to train a simple feedforward neural network to learn preferences over the sequential data. We evaluate the impact of active learning on the number of traces that are needed to train a model that is accurate and interpretable. This evaluation is done by comparing the aforementioned feedforward network to a more complex neural network model that uses LSTMs and is trained with a larger dataset without active learning.

## Introduction

When we have a human-in-the-loop during planning, learning that person's preferences over plan traces becomes an important problem. These preferences can be used to choose a plan from amongst a set of plans that are comparable by the planner's cost metrics. Such a plan would naturally be more desired by the human. The user may not like to constantly dictate their preferences, and may not always be in the loop during execution. Thus, it is important for the user's preference function to be learned well, and for the user to be able to verify them. For verification, there ought to be a way to interpret how the model's decisions were made, and verify how faithful the learned model is to the user's preferences.

A user's preferences function may be quite complex with dependencies over different subsets of features. The utility of some features maybe non-linear as well. Such a preference function may require a fair amount of information to approximate. We cannot expect a single user to give feedback over a large set of traces to get the relevant information. So Active learning, with a sufficiently expressive user interface for feedback, is essential to minimize queries and redundant information.

In this work, our objective was to model the user's preferences over plan traces. There do exist techniques that

efficiently represent and reason about preference relationships. CP-nets (Boutilier et al. 2004) and Generalized additive independence(Braziunas and Boutilier 2006) models are typically used to represent preferences over sets of variables without consideration to the order in which they appear. While these models can be adapted to handle sequential data, they are not intended for it. LTL rules, however, can capture trajectory preferences very well and are used in PDDL 3.0 (Gerevini and Long 2005), and LPP (Bienvenu, Fritz, and McIlraith 2006). However, it can be very hard for a user to express their preferences in this form. We discuss existing approaches in more detail and the differences with respect to our work under the related work section.

In our approach to learning preferences, we want to efficiently identify the relevant features and the degree to which they affect the preference score of a plan. We thus employ a feature-directed active learning approach that specifically picks plan traces that are most informative about the feature's effects on preference. After active learning, we encode a plan trace in terms of the relevant features it contains. We gather a set of training data from active learning, along with the user's preference score to help train a simple Neural Network (NN) that we call the *FeatureNN* model. We use a Neural Network as they can approximate complex functions to a good degree. Our approach is in one way, related to Generalized Additive Independence in that we try to learn a utility function over pertinent features, but we do not explicitly define or restrict the form of any utility functions. Rather a simple one hidden-layer feed-forward neural network learns the functions, dependencies, and relative weights over the relevant features. The *FeatureNN* then predicts a preference score for each plan reflecting the user's preferences. We also compare the performance of the *FeatureNN* to another *SequenceNN* model that processes sequential data using an LSTM(Schmidhuber and Hochreiter 1997) module. The *SequenceNN* is not trained with data from active learning, but with a larger dataset of traces with ratings. This is to evaluate how efficient our active learning approach is with respect to the number of traces. Specifically, we compare the number of traces required by *SequenceNN* and *FeatureNN* for the same accuracy and interpretability.

Neural networks, unlike linear functions, are not as easy

to interpret. Even simple NN with a single hidden layer can be a challenge. We help the user interpret the decisions of the neural network by showing how the preference score is affected by removing different features of a plan trace. This is similar to using Saliency Maps (Simonyan, Vedaldi, and Zisserman 2013) in images to explain what parts of the image contributed to the classification. In this way, we can explain to the user what plan features contributed to the preference value and by how much. The difference in preference score should correspond to the user's expectations as per their preference model. The more similar the effect of changes are to the user's preferences, the more interpretable the NN model is to the user as it approximates well their own preference function. Such a method of explaining a decision(score) is also related to explaining using counterfactuals (Miller 2018). Here the counterfactual is the plan trace without a specific feature. Additionally, when the specific features used to compute preferences comes from the user's feedback (during active learning), this interpretability is obviously improved.

We present our work by first defining the problem before delving into the methodology of our approach. In the Methodology section, we discuss the domain used, the user preference model, and the feature-directed active learning process. We also discuss the two neural network models used to learn the preference model, viz. the *FeatureNN* and the *SequenceNN* models. Then we present our experimental results in which we compare the two models with respect to their accuracy in predicting the preference score, as well as interpretability. Lastly, we discuss the results and possible extensions to the work.

## Problem Definition

Given a Domain $D$, with a set of features $F$, and a planner $P$, the problem is to learn the preference function $F_p()$ that captures the user's preference model $U_p()$ and scores traces accordingly. The types of preferences we learn in this work are a function of the feature set $F$ of the domain and not hidden variables or action costs. The user $U$ is available to rate a plan trace on its preference, and annotate what features contributed positively or negatively to the rating. Features can be categorical or cardinal(count), and involve sequences.

Plans are rated between $[0, 1]$ with higher values indicating a greater preference. If there are no features in the plan that either contributed positively or negatively to the preference, then the preference score is $0.5$.

An equivalent problem formulation assumes that instead of a domain D and planner P, we are given as input a large enough set of plan traces $B$ (backlog of traces) over a relevant set of initial and goal states. We assume that this set of plans covers the space of possible preferences that the user might have.

## Methodology

For our experiments, we chose to use gridworld with features that any human can relate to. We chose gridworld as it is easy to quickly generate many diverse plans that cover the range of features.

Given the domain and a task, we go through $r$ rounds of active learning. Each round comprises of $r_t$ traces. Both $r$ and $r_t$ are hyperparameters. For our experiments, we set the number of rounds at $r = 3$. After acquiring the data, we train the NN model and test it on a hidden set of traces. We now go over different parts of our methodology in detail.

## Domain

The objective in our gridworld domain, which we call $Journey - World$, is to travel from home to the campsite which is shown in the grid in Figure 1(a). Each step of a plan corresponds to a cell of the grid. While some cells are empty, there a lot of cells that have features. These features can be eateries(like a coffee shop, restaurant), landmarks(natural history museum) or activities(visiting the library, watching a movie). The user can move through any of these states before reaching the campsite. A majority of the cells also contain landscapes like mountains, lake, sea or industries. The user is not allowed to move through $Landscape$ cells. Moving through cells adjacent to $Landscape$ cells corresponds to seeing the landscape along the journey. For example, if a step in the plan goes through a cell which is adjacent to a lake, this corresponds to the plan going through a state where the user passed by a lake.

All non-landscape features (like coffee, donut) are binary features in a plan trace i.e. the user has either visited one or not. On the other hand, the landscape features are cardinal, i.e. we count the number of such landscape features in the plan trace. We assume that the count of cardinal features can make a difference in the preference score. In total, there are 13 features in $Journey - World$.

We had designed $Journey - World$ with simple and commonly understood features to make it easier for subsequent human-studies. We assume people will have preferences over these features.

## User Preference Model

For our current experiments, we chose to use a completely defined user preference model to represent the user. This made it easier for us to test and debug our methodology. In future extensions of this work, we will include evaluations with human trials. The user's preference model is defined as follows

$$P(trace) = \begin{cases} 0.5 + 0.1 * (C) - 0.3 * (D) + 0.1 * n(L) \\ +0.1 * n(I) \qquad if\ not(C\ and\ D) \\ 0.5 + 0.3 * CD + 0.1 * n(L) + 0.1 * n(I) \\ \qquad\qquad if\ (C\ and\ D) \end{cases}$$

$where\ n(x) = min(x, 2)\ , C \in \{0, 1\},\ D \in \{0, 1\}$
$and\ L, I \in \mathbb{N}$

$C$ is a binary variable that is 1 when the plan trace has a coffee. $D$ is also a binary variable and represents a Donut. $CD$ represents a binary variable set to true when the plan trace a coffee and a donut. When $CD$ is true, $C$ and $D$ are false and this dependency affects the preference score computation as shown in the preceding equation. $L$ and $I$ represent

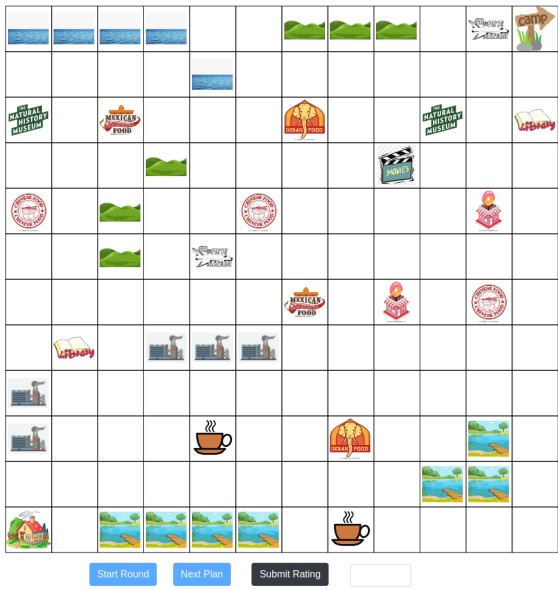

(a) Problem domain

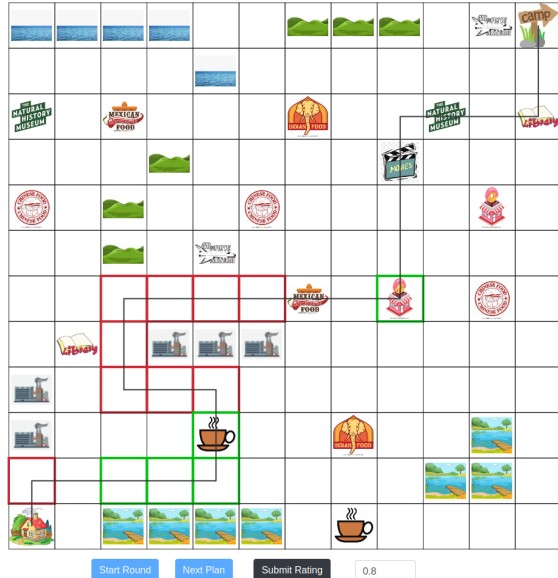

(b) Rated and annotated plan trace

Figure 1: (a) Problem domain. The task is to go from home(lower left corner) to the camp site(upper right corner) (b) A plan trace that has been rated $0.8$ by the user. The user has also provided annotations: green for liked features, and red for disliked features.

the number of lake and industry regions respectively. These are cardinal features, and the preference of the plan increases based on their count, up to 2, and then stops increasing. The function itself, while simple to understand, is non-linear and hidden underneath a large hypothesis space of functions that could be learned in the domain, over a larger set of features (13 in total for the $Journey - World$ domain). In our experiments, we programmed a separate module to rate and annotate plan traces based on an input preference function like the one described previously. This synthetic human is what rates and annotates in the active learning process that we will describe shortly. Using a synthetic human helped speed up the testing and debugging process, and gives us a baseline noiseless scenario to test against.

## User Interface

The current user interface(Figure 1(a)) for $Journey - World$ displays the entire grid. Icons are used to show the features present in cells. The plans for each round of active learning are then shown one at a time. The plan steps are visualized as a line going from the home to the campsite. The user has to input a rating to indicate their preference for the plan based on the features that are visited. They can also annotate features of the plan that they like(green) or dislike(red) as shown in Figure 1(b). The user can click on the $Next\ Plan$ button to then move on to the next plan. The interface automatically switches to the next round of Active Learning when the current round's plans have all been rated.

## Feature-Directed Active Learning

In our active learning process, we go through multiple ($r$) rounds of feedback. Each successive round utilizes the knowledge from previous rounds to select the most informative queries. In the first round, the user is shown the most diverse set of plan trajectories that were generated for the domain. We choose diverse plans because in the first round we do not have any knowledge of what features might affect the user's preference and hence, we want to cover the feature space as much as possible in $r_t = 30$ traces. In order to get the required diverse plan set, we first generate a large number of plan traces(10000 plans) over a user specified set of initial and goal states that we refer to as the backlog of plans, $B$. In our current experiments we only have one initial and one goal state. We are easily able to generate such a large backlog of plans because it is a type of gridworld domain. We did not want the computational cost of diverse plan generation to hamper the work. This is a computational cost that needs to be considered when working with other domains. The plans generated cover the entire feature space of the $Journey - World$ domain. We then select the 30 most diverse plans within the set of backlog plans for the first round. We will now discuss how this is determined.

The diversity score between any two plans $p_a$ and $p_b$ is denoted by $d(p_a, p_b)$. The diversity is based on the sum of feature count differences for features $f \in F$ that are present $p_a$ and $p_b$. For a particular feature $f$, we compute the feature count difference $f_\Delta$. Rather than use the difference in count per feature, we use a geometric series sum as computed in Equation 1. The first count in the difference contributes 1,

the second count contributes a 0.5, the third contributes 0.25 and so forth. So the count difference for a single feature contributes to at most 2 to the diversity computation. This avoids any single feature from dominating the diversity computation.

The diversity between two traces is computed as the average $f_{E\Delta}$(Equation 2) over all the features in the domain. Finally, we calculate the backlog-diversity $d^B$ for a plan $p$ using equation 3. The backlog-diversity is the average pairwise diversity over every other plan in the backlog. Using this diversity score, we select the top $r_t$ plans ($r_t = 30$ in our experiments) for the user feedback.

$$f_{E\Delta} = \frac{(1 - r^{f\Delta})}{1 - r} \qquad (1)$$

$$d(p_a, p_b) = \sum_{f \in F} f_{E\Delta}/|F| \qquad (2)$$

$$d^B(p) = \frac{\sum_{p' \in B} d(p, p')}{|B|} \qquad (3)$$

After the first round of diverse plans, we then make use of the ratings and annotations provided by the user in the first round to generate the most informative plan traces for the subsequent rounds. Given our acquired knowledge of relevant features from the previous round, our objective now is to figure out the effects and dependencies between these features. We also want to select traces for the next round that are more likely to be rated either significantly higher or lower. This region of data is typically harder to get as we expect most data to be closer to the average. In order to estimate which plan traces would be either most preferred or least preferred, we use a fast $weak\_predictor$ that predicts the rating of any arbitrary plan $p$ given prior knowledge. We need the predictor to be fast as we have to give traces or queries for the next round in a short amount of time.

The $weak\_predictor$ estimates a value for each feature based on the prior annotated data. It can then estimate the score of an unrated plan as just the sum of the features present in it. The value of each feature is scored using a quick and simple method. First, for each scored plan trace $p$ with rating $r_p$, the feature $f$ is given a score $f_{score}^p$ for that plan by equation 4. Then the feature's score, $f_{score}$, is computed as the average $f_{score}^p$ over all plans that the feature appears in. Then to predict the score for an unrated plan, the $weak\_predictor$ assigns a score $predict(p)$ which is the sum of $f_{score}$ for all features present in the plan.

$$f_{score}^p = \begin{cases} r_p & if\ f\ annotated\ as\ "liked" \\ -(1 - r_p) & if\ f\ annotated\ as\ "disliked" \end{cases} \qquad (4)$$

In addition wanting plans that are likely to be rated much higher or much lower, we also want the next round traces to have two more properties. We still want to include some diversity in the plan traces with respect to the overall backlog of traces to uncover features that we might have missed in the first round. Additionally, we want to maintain some similarity in traces between the rounds. We think that the

similarity between plans reduces the cognitive load on the user as they need not parse wholly different traces. Given a plan $p$, we denote its similarity to the already scored traces as $S(p)$.

Finally, we assign a combined weighted score of $p_c$ to all the plans in the backlog given by equation 5. The top $r_t = 30$ plans are then picked for the next round, and in this way the active learning proceeds for $r$ rounds. For our experiments $r$ is 3 rounds.

$$p_c = w_1 * predict(p) + w_2 * d^B(p) + w_3 * S(p) \qquad (5)$$

## Preference Learning using Neural Networks

For learning the preference function we used two models, *SequenceNN* model and *FeatureNN* model.

The *SequenceNN* model uses an LSTM module in it. We considered an LSTM based model as they are well suited to learning patterns overs sequential data. The input plan trace was encoded such that each step was an encoding over the features of the cell visited at that step. There are 13 features in total, and so each step is a 13-dimensional vector. We do *not* provide or restrict the input to only the features that the user annotated during active learning for the *SequenceNN* model. We wanted to test how easily the model could still figure out the relevant features and learn the preference function well.

The training data for the *SequenceNN* model was a set of rated plan traces. We varied the number of traces given from as small as 30 to 12,000 in increasingly larger step sizes. A plan trace would be a $N \times 13$ array where $N$ is the plan length. We trained the model for 10 epochs with a batch size of 8, a learning rate of 0.01 and using stochastic gradient descent. The *SequenceNN* module in our model has 16 memory cells. After processing the plan trace through the LSTM module, we concatenate the output vector and memory nodes of the LSTM module and pass it through a single fully connected hidden layer, followed by the output layer which outputs the preference score between [0,1]. The model summary is in Figure 2. The idea is that the LSTM module output and memory, at the end of processing the sequence, will have the necessary information related to the sequence for predicting the score.

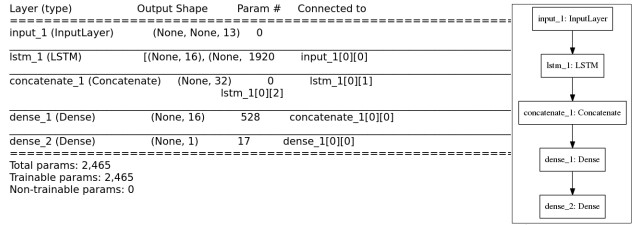

Figure 2: LSTM Model for Preferences

To test if any variant of the *SequenceNN* model could learn the relevant information better, we also tried varying the size of the model (number of parameters) to make it more powerful. We varied the model from 16 to 64 weights in the

memory layer. Results are in the Evaluation and Analysis section.

For the *FeatureNN* model, the input was an encoding of the plan trace that only comprised of the features the user annotated as relevant during active learning. The entire plan trace was summarized into one encoded vector. For example, in the user preference model in our experiments, only 5 features matter to the user. We determine what these features are through active learning, and then defined our *FeatureNN* model accordingly to take a 5-dimensional vector as input. For example, if a plan trace had a step with coffee and steps that passed by 3 lakes, then the values at the corresponding indices are set to 1, and 3 respectively. Note that since coffee is a binary feature, even if two coffee steps were in the plan, it's value in the encoding is only either 1 or 0. As for the model description of *FeatureNN*, it was a simple fully connected neural net with one hidden layer of 4 dimensions and one output layer. The model summary is in Figure 3. Note that 4 dimensions or nodes for the hidden layer is not a magic number, and would need to be larger if there were more features. We reduced the number of dimensions for the hidden layer until the results were measurably worse. To train the *FeatureNN* model we vary the number of traces per round $r_t$ from 5 to 50 traces for $r = 3$ rounds. Since the dataset size is very small (smallest is 15 traces), we create 200 duplicates of the data points uniformly and train for 10 epochs. We also shuffle the data and train with a batch size of 8, a learning rate of 0.01, and using stochastic gradient descent.

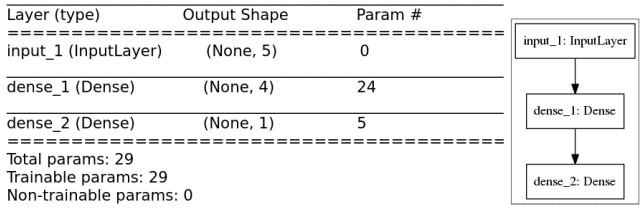

Figure 3: Simple NN Model for Preferences

Both models were tested on an unused test set of 1000 traces for accuracy and interpretability.

## Interpretability of the Preference Model

A user can interpret a Neural Network's behavior through analyzing what features are salient to its decision, and by how much. This can be analyzed by adding or removing features and seeing the resultant effect on the predicted score. When done over a set of different traces, the user can intuit what mattered and how much. With this in mind, we compute a measure of interpretability we call the Attribution Error $AE$. The $AE$ for a feature $f$ of a Plan $p$ is computed as follows:

$$AE(p, f) = |(U_p(Plan) - U_p(Plan - f)) - (F_p(Plan) - F_p(Plan - f))| \quad (6)$$

where $U_p()$ is the preference function of the user (true model of preferences) and $F_p()$ is the learned preference function. $AE$ is simply the difference in the effect of the feature on the preference scores. The overall $AE$ for each test plan $p$, $AE(p)$, is the average of $AE(p, f)$ for all $f$ present in $p$. We compute the $AE$ score for the test set as the average over only the top 10% of AE(p) errors. We do this because neural networks can sometimes have enough capacity to memorize many cases and increase accuracy. So it can predict the correct preference score of both of the original trace and modified trace (with dropped feature) by the memory of very similar traces. It would then seem like it's preference function predicts the same way as the ground-truth preference function, but it maybe using unrelated features. Therefore, it is in the failure cases that we get a true measure of its generalization and how faithful it is to the true model of preferences. That is why we use the average over the top 10% of AE(p) errors. These failure cases could correspond to the cases when a rare or unseen pattern of features are input, and thus not memorized.

## Evaluation and Analysis

### Evaluation of LSTM model

When varying the number of training input traces given to the *SequenceNN* model, we observed that the accuracy improved (error decreased) as expected(Figure 4(a)). Surprisingly, even with 30 traces, it was able to predict with an error of 2.5% over the test set of unseen 1000 traces. We attribute this to the fact that there are enough simple correlations with other features that can predict the score well for the preference function that we tested with. This is evidenced by the fact that the interpretability measure (Attribution Error) is very low for 30 traces(Figure 5 (a)); The attribution error was greater than 0.3 and the value range of $AE$ is [0,1]. Additionally, we give the most diverse $N$ traces for each training set size to the *SequenceNN* model. Diverse traces are more likely to contain relevant information.

The interpretability of the LSTM model was not impressive. The attribution error did decrease over the range of training set sizes, but only as low as 0.09 as shown in Figure 5(a). Given that the preference scores are between $[0, 1]$, this would correspond to a 9 percent error after 7500 rated traces. Needless to say, it is unreasonable to expect a single human to rate 7500 traces.

We also tried varying the size of the *SequenceNN* model from 16 to 64 dimensions. This improved accuracy by a minuscule amount (order of $1e-4$), and interpretability did not improve.

### Evaluation of Feature-NN model

The performance of the *FeatureNN* model was significantly better both in accuracy(lower error) and interpretability than the *SequenceNN* model as seen in Figure 5. This should come as no surprise since we restrict the input space based on user feedback (knowledge) on relevant features. This also restricts the hypothesis space of functions that the simple feed-forward network could search over. We think this will make it more likely that the NN will find a good and faithful

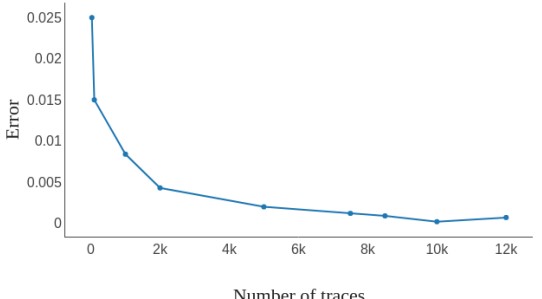

(a) Error using rated traces and a LSTM based learner

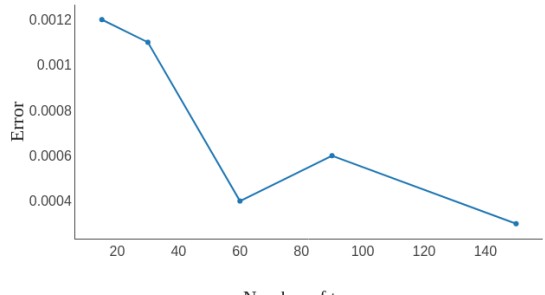

(b) Error with feature-directed active learning and a simple neural network

Figure 4: Comparison of accuracy

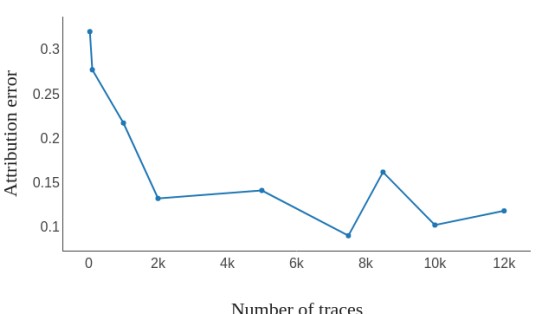

(a) Attribution error using rated traces and a LSTM based learner

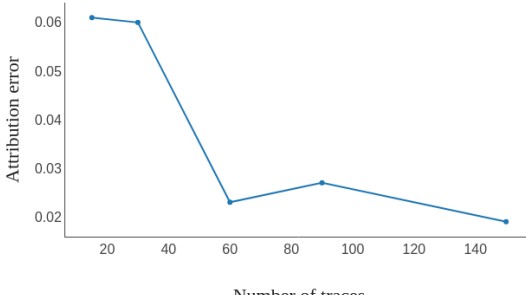

(b) Attribution error with feature-directed active learning and a simple neural network

Figure 5: Comparison of attribution error

approximation function to the true preference function. This is as opposed to discovering predictive but incorrect correlations.

What is interesting to note is that the interpretability, as measured by the $AE$ error drops to as little as 2.5 % in as little as 60 traces (20 traces per round over 3 rounds) for *FeatureNN* model as shown in Figure 5b. It drops below 2 % with 150 traces.

The *FeatureNN* model with 90 traces is as accurate as the *SequenceNN* with 7500 traces in our problem, with 8% less Attribution Error (more interpretable). While we expected *FeatureNN* to be better, we did not expect such a large difference in efficiency.

## Analysis and Discussion

Even with as little as 13 features and a relatively uncomplicated preference function, a sufficiently powerful *SequenceNN* model did not find the underlying preference function. Instead, it found correlations that predicted the preference score to a very high level of accuracy. This, unfortunately, makes the model suffer in interpretability.

As the number of features increases, the hypothesis space of a NN will increase significantly. This makes it much more likely for any NN to find spurious correlations, and suffer in interpretability. So active learning and using a simpler NN becomes very important for learning preferences in plan traces.

As for prior feature knowledge, we assumed knowledge about what features were categorical (binary in our experiments) and what features were cardinal. Rather than assume this knowledge, we can get this from the user as well, and reduce the assumptions about the domain features. Alternatively, we could have just encoded all features as cardinal features, and let the neural network determine what features were categorical. While this is certainly possible, we think it better to get this knowledge from the user and encode the plan trace based on this knowledge. This makes the job of the neural network easier, and less likely to learn spurious correlations.

In our current encoding of features in *FeatureNN* model and our experiments, we have not included a preference dependency that considers the number of steps between fea-

tures. For example, *I would like to have a donut within 3 plan steps after having a coffee*. This omission was not intentional. One can easily encode such a sequential feature as a variable as well. The number of steps between the two (state) features becomes a cardinal variable to represents this sequential feature.

## Related Work

Two well known paradigms for learning, representing and reasoning over preferences are CP-nets and Generalized additive independence (and their variants). Both of them were intended for preferences over outcomes. Each outcome can be comprised of many parts(decisions). One can think of each decision as choosing a value for a variable. The user would have preferences over the possible outcomes, or there maybe a utility (value) associated to each outcome. In CP-nets (Boutilier et al. 2004), the decisions or variables are represented in a graph, and there exists dependencies over variables. The preferences of a variable's values are affected by the value of the parent variables. The CP in CP-nets stands for *Ceteris Paribus* or *"all else being equal"*. Here, the *all else* refers to the parents of the node, and when they are equal, then a particular set of preference orderings for the child variable's values hold. The knowledge of the dependence graph is either known apriori, or can be queried from the user. Once the hierarchy of dependence is known, the user is then queried about preferences at each node. For CP-nets to be used in plan trace preferences, we would have to ask the user what the dependencies are. Then we would have to ask the user for their relative preferences over features, given fixed parent feature values. Note that the variables for plan preferences may also have to incorporate information about order. So there are significantly more variables (features) to consider in sequential data versus unordered data. We think querying for such knowledge is very demanding, and not natural for preferences over plan traces.

We believe it is more natural for the user to specify the relevant conditional dependencies over features while annotating a plan trace. Additionally, we think it easier to give a preference value for the plan trace rather than relative preference orderings over the features in the domain. The features could include sequential dependencies or position-dependent features. We think it would be hard for the user to be able to describe sequential features and the relative ordering over them. Lastly, CP-nets do not compute utility values and some outcomes can be incomparable for a particular network of dependencies. In our problem, we would like a total order over the plans, to select the most preferred plan. So it helps to have a utility/preference value for every plan trace.

On the other hand, GAI (Braziunas and Boutilier 2006) models do provide a single utility value for a set of features. As stated in their work, they provide an additive decomposition of a utility function (into sub-utility functions) in situations where single attributes are not additively independent, but (possibly overlapping) subsets of attributes are. Since the subsets of attributes for the different sub-utility functions can overlap, one must query either with only global queries or a combination of local queries (over the subsets of features) with global queries to calibrate (as was done in the GAI work) (Braziunas and Boutilier 2006)(p. 3). To learn a GAI with active learning from a single user, there are one of two methods. We could make assumptions about what subsets of variables are part of each sub-utility function, and what those functions are, or the user would need to know and give us this information. We think this is a very difficult task for the user. In our approach, we only do full trace queries and ask for annotations and preference ratings. We think it is more natural to ask the user for their overall rating of a plan, rather than how much each subset of features affected the rating. The neural network then handles the job of learning the preference function over the user-specified features (and any dependencies).

The other formalism for specifying preferences are LTL rules (Huth and Ryan 2004) (p.175), which allow the user to specify sequential patterns. Expecting the user to be able to specify LTL rules might be unreasonable. The user would also have to give utilities or preference orderings over the specified LTL rules. One can interpret our interface as extracting a subset of simple LTL rules (through annotations) which are present in a plan trace. The user gives a rating to the trace, as well as what features (LTL rules) were good or bad. Extending the LTL analogy, our encoding of a plan trace can be seen as a vector of the relevant LTL rules. The index corresponding to an LTL rule is set to 1 if the rule is satisfied in the plan trace. However, recall that we also allow cardinal features (counts) in our encoding, and not just binary variables. Our interface and learning framework does not handle the entire gamut of possible LTL rules. We are working on extending the types of sequential preferences supported, while keeping the interface intuitive and expressive.

## Conclusion and Future Work

In our approach, we use feature-directed Active Learning complemented with an intuitive and expressive user interface to learn the user's preference function efficiently. The traces obtained during active learning are rated and annotated by the user. These traces are encoded as a vector over the features that the user indicated as relevant to their preferences. The feature vectors are used to train a simple feedforward Neural Network to learn the preference function. We show that the *SimpleNN* neural network is more accurate and interpretable with fewer, more informative plan traces as compared to the LSTM based *SequenceNN* model. The latter was trained with a larger dataset of rated plan traces without active learning.

Our current experiments use a user preference function over only a few variables. It is important to see how efficiently our framework learns a more complex preference function. Moreover, the current preference function is completely deterministic as it provides consistent annotation and rating to the plan trace. A human, however, might not behave in a consistent manner. We will test with a noisy or probabilistic preference model in future work.

The user interface itself can be extended to include more complex annotations. For example, the user can also provide annotations for some features to be added/dropped from the plan. This is especially useful for cardinal feature as the

modified feature count represents what is ideal to the user. For example, if the user's preference doesn't increase after visiting more than 2 *lakes*. Then this can be communicated by removing extra *lake* features from a plan trace.

We have mentioned categorical and cardinal features, but our framework is also intended to support real-valued features. We would need to adapt our active learning process to elicit feedback as to what the minimum, optimum and maximum values of such features are. These would be the minimum essential points to sample for approximating the underlying utility function.

Lastly, we would like to simplify the function by which we choose plan traces in successive rounds of active learning. We think that the similarity with traces from previous rounds is unnecessary, and might not appreciably reduce the cognitive load on the user. We think that just diversity and selecting traces that are much more preferred(closer to 1.0) or much less preferred(closer to 0.0) would be sufficient.

## Acknowledgments

This research is supported in part by the ONR grants N00014-16-1-2892, N00014-18-1-2442, N00014-18-1-2840, the AFOSR grant FA9550-18-1-0067, NASA grant NNX17AD06G and JP Morgan faculty research grant.

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
