# OpenReview forum: "Feature-directed Active Learning for Learning User Preferences"
_icaps-conference.org/ICAPS/2019/Workshop/XAIP — XAIP 2019_

### Official Review · AnonReviewer1 · 2019-05-09

**Rating:** 4
**Confidence:** 2

**Review:**

Authors propose an approach for learning user preferences through active learning scheme. The model is built such it can identify relevant features of the preferences and how they impact the plan. For this purpose, the authors provide a measure of interpretability.

The paper is well structured and easy to follow. Authors provide an example problem and evaluation of the proposed approach. Since the user preferences are captured with LSTM and Feature-NN models, authors could extend the discussion in related work to other approaches used from explaining and interpreting decisions of a neural network.


More suggestions and questions:
- Models in figures 2 and 3 could be better represented.
- page 5: 9 percent -> 9%
- The main problem for getting interpretability measure of a model in a real-world scenario would be obtaining ratings from humans. What is the minimum meaningful number of samples?
- The error rate in figure 5 has some changes for the number of traces more than 8k. Is this due to  an overfitting problem? Maybe more tests with a different number of traces (more granular) or doing cross-fold tests would reveal more information.
- Tests with real users would be useful in the future to show the effectiveness of the method as well as to get user feedback on informativeness and interpretability of the process.

---

> ### Author Response · Authors · 2019-05-14
> **Response**
>
> Thank you very much for reviewing our paper and for your feedback. We will make the edits suggested. Regarding some of your concerns:
>
> 1. What is the minimum meaningful number of samples?
>
> This would depend on the complexity of the user’s preference function. We can give a lower bound estimate using the following simplified compilation; we can consider each plan trace and preference value as a linear equation with weights over features, then we have a system of linear equations. So the minimum number of traces needed would be equal to the number of features affecting the user’s preference function. This is assuming no non-linear, or conditional interactions between the features and so is a lower bound.
> In future work, we hope to query for samples based on the uncertainty of the model, and so we will be using Bayesian model learning techniques.
>
> 2. Increase in error rate after 8k traces in LSTM:
> 	We attribute this spike in error to overfitting (as you pointed out). We think this is connected to the problem of spurious correlations in the sequential data that the LSTM model seems to learn. The simpler feed-forward NN, while not immune to this problem, was less susceptible to it as there are fewer features to learn correlations over.
>
> 3. Testing with real users:
> We plan to do human trials after adopting Bayesian model learning to incorporate uncertainty based sampling of plans. We agree that testing with humans will be a good measure of effectiveness of our methodology.

---

### Official Review · AnonReviewer4 · 2019-05-13
**preference models via "interpretable" NN**

**Rating:** 3
**Confidence:** 2

**Review:**

This is a nicely written paper proposing to learn NN representing user preferences. They introduce a methodology comprising first a feature-identification then a NN training step. They test their method on a grid-world example against a naive NN method.

I'm not particularly happy with the comparison to the NN. It seems rather artificial. The features in the example are all already defined on traces, so what is the point of feeding any representation ior learning mechanism with a sequence thereof? It's Ok fior a workshop, but in future iterations I would recommend to think carefully about comparison fairness.

Overall, as a preliminary effort using NN to represent user preferences the paper seems worthwhile to me.

I'm rather non-plussed though by the notion of "interpretability" considered here, which seems to be the main link to XAIP. The notion measures the error of the NN in the preference difference stemming from removing a feature. Even when accepting that feature-removal is a good means to explain a user preference, I don't see how "interpretability" measures the ease or difficult of doing so. It seems much more a measure of learning accuracy.

Adding to this that NN, in comparison to explicit measures like CP-nets, are by their very nature opaque, it seems to me more than doubtful that this paper represents progress, rather than a step back, in explainable user preference representation and management. The point about lack of support for temporal behaviour in CP-nets is also moot as the proposed approach hasd the exact same lack of support. Instead of learning an NN over the advertised "features", one could just as well write down a CP-net over these same features. For inherent temporal support in an NN, an LSTM or similar would be required, which is what the authors are proposing an alternative to.

In any case, for discussion atb the workshop the work certainly is interesting so I am not opposed to accepting it.

Minor: I don't understand how "When CD is true, C and D are false". CD s the conjunction of C and D no?

---

> ### Author Response · Authors · 2019-05-14
> **Response Part-1**
>
> Thank you very much for taking a detailed critical look at our paper and for your feedback. We have some responses to your review.
>
> 1. The issue of what is “CD”:
> In our example in the paper, CD is the conjunction. We do support ordered features as well, such as C->D (C followed by D).
>
> When we encounter C and D in a trace. We set CD to 1, and C to be 0 as well as D to be 0. Now instead of learning the fact that C and D are non-linearly related, the NN can simply learn the effect of CD which we conjecture requires less number of traces. We also found this to be empirically true.
>
> 2. Regarding the use of features and comparison with CP-net:
>
> CP-nets induces a partial ordering over the instantiation of all features. Hence, not all instances are comparable with CP-nets, that is one of the known limitations. This is where Generalized Additive Independence (GAI) models fit into the literature. These assume that there are subsets of variables, whose utilities are independent of other subsets and their individual (subset) utilities can be summed up to get the total utility. The utility of each subset is a function over the features in that subset. For example, if there were 3 features, the preference function could be
> P = f_1(A,B) + f_2(C).
> We could determine what those subsets are by querying, and then determine or assume what function or dependencies exist within each subset. This would require a lot of queries to identify the subsets and the utility function for each.
>
> Our feature-based active learning approach tries to learn the GAI function with  fewer amount of queries. The initial diverse set of plans present a good coverage of the possible features. This allows the user to tell us which features really matter for their preferences. After that, our plan rating heuristic further selects combinations of already annotated features that would potentially also be relevant to the user. We believe this would lead to learning the preference function more accurately with fewer traces.
>
> The use of NN is actually important for the case where the user doesn’t annotate a subset of features even though it impacts the preference rating. The neural network can still figure out that the dependencies between features that are relevant to the user’s preference.
>
>
> 3. Our notion of Interpretability:
> The notion of interpretability that you state is the “ease of explanations”. A different notion of interpretability is “the degree to which a human can consistently predict the model’s result” as stated in the work of Been K. , Khanna R., and Koyejo R. (http://papers.nips.cc/paper/6300-examples-are-not-enough-learn-to-criticize-criticism-for-interpretability). By dropping or changing features, if the model’s prediction match what the human would predict based on their own model of preferences, then it is more interpretable. Yes we can see it as another measure of accuracy as it is like evaluating the model on another data point. It also serves to test how the model is sensitive to features (at least locally) and if that sensitivity matches how the user’s own preferences would change, then it is more predictable to the user.

---

> > ### Author Response · Authors · 2019-05-14
> > **Response Part-2**
> >
> > 4. Regarding the use of Simple feedforward NN vs LSTM:
> >
> > During our initial experiments with using LSTMs we noticed that the LSTMs tended to learn a lot of spurious correlations. For example, if highly preferred traces happened to pass through a particular subsequence of features, then that would be used to score subsequent traces. This subsequence of features was not annotated by the user, and just happened to be a correlation that existed in the data. LSTMs seem more likely to use such spurious correlations, especially if they occur towards the end of a sequence. We tried a data augmentation approach that varied the number of steps and irrelevant features while keeping the same plan rating for a given data point. Despite a significant amount of data augmentation, the effect of spurious correlations was still significant.
> >
> > So we took a different route. Rather than the LSTM learning to represent the plan trace in the last step, why not represent the plan trace by the state and sequence features that the user annotated. For example, since the user tells us that they prefer CD (Coffee followed by a Donut), if the sequence CD appears in a plan trace, then the vector representation corresponding to that plan trace would have CD set to 1. So we encode the entire plan based on the annotated features that appear in it.
> >
> > The implicit assumption here is that there is a function over these features that approximates the preference function well. That is to say, there are no further sequences or orderings of those annotated features that would significantly affect the preference. We assume that all the salient orderings have been annotated and become features before we train our NN model. To make sure of this, we use a diversity measure to increase the coverage of possible features throughout the interaction with user.

---

> > > ### Comment · AnonReviewer4 · 2019-05-14
> > > **Thanks for the feedback**
> > >
> > > Thanks!
> > >
> > > I'm still not really convinced of the notion of interpretability here -- agreeing with the system's predictions coukd instill trust I suppose. Is this the same as "understanding"? Not in my mind. But happy to discuss this at the workshop.

---

### Official Review · AnonReviewer2 · 2019-05-14
**Interesting paper, ML part might be enriched**

**Rating:** 2
**Confidence:** 2

**Review:**

The authors propose a NN-based model to learn user preferences over plan traces. The goal is to identify the relevant features and the degree to which they affect the preference score of a plan.
The paper is well written though some parts might be improved, e.g, introduction is too dense and do not clearly declare the contribution of the paper. It is unclear to me the contribution of the proposed approach with respect to the state of the art.

COMMENTS:
-- Given the nature of the paper, that employs ML algorithms, I was expecting to see more on the ML side in terms of ML algorithm used (SVM with different kernels, RF, DTs, etc.), such as:
	-- Why did you select NNs as they are - as noted - not easy to be explained at all?
	-- Why you do not provide accuracy values about the learning process?
	-- did you use a grid-search to fine-tune parameters?
	-- did you use a training/test/validation phase?
	-- 1000 plans are few for NN. Why you do not apply resampling to improve the training dataset? Is the dataset balanced?

-- The authors propose feature relevances as a way to explain the model prediction. In my view, feature relevances do not represent an explanation by themselves, as they do not encode the causality, that is a key element of explanations. I think the authors might plan to employ global interpretable models to explain the system (see, e.g., Guidotti, Riccardo, et al. "A survey of methods for explaining black box models." ACM computing surveys (CSUR) 51.5 (2018): 93.)
-- Not clear to me what is the meaning of "error" in the experimental evaluation. As you are evaluating a predictor, you might rely on classical metrics such as F1-score at least. Here here seems to encode only true-positive...


MINOR:
-- The "Methodology" sections starts with "In our experiments....". Furthermore, a graphical overview of the approach might be beneficial here.
-- A working example might be useful in the introduction to better present the problem

---

> ### Author Response · Authors · 2019-05-14
> **Response to Review**
>
> Thank you for taking a look at our paper.
> Our main contribution is proposing a Feature-Based Active Learning approach to learn an interpretable model of the human’s preferences. By extracting the relevant features and using them to define the model input, we expect the model to be more interpretable. The choice of the model to learn the utility function on the extracted features, is another dimension that can affect the interpretability.
> By using only those features annotated by the user, we improve the interpretability of the neural network, which is an opaque model. We chose to use a neural network, as the function learned in the last layer of a NN matches a generalized additive independence (GAI) model of utility, which is what we wanted to learn. The GAI model of utility by Braziunas.D and Boutilier  (https://www.aaai.org/Papers/AAAI/2006/AAAI06-253.pdf) is the paradigm typically applied for learning utility functions in preference elicitation. GAI assumes that subsets of features contribute a utility amount towards the total utility which is the additive sum over these subset utilities. There exists a function over each subset that determines the utility value from it.
>
>
> 1) We didn’t use F-1 scores as that is a metric suited for classification problems, and our work is on a regression problem; we are trying to learn a utility function that maps annotated features to the preference value. On a similar note, SVMs are typically used for classification.
>
> 2) Random forests and Decision Trees can be applied for regression. We agree that they are more interpretable when there exists some hierarchy over the set of features.  Our reason for not considering it over a simple feedforward neural network is that we do not assume or expect there to be a hierarchy over the features. Based on the GAI model that we mentioned earlier, subsets of features contribute a utility value based on some function over each subset. These subset utilities add up to give the overall utility (preference). There is no notion of a hierarchy over all features in the GAI model, and the decision tree would try to learn a hierarchy. We think this would confuse the user when analyzing/interpreting such a model, as the hierarchy is arbitrary ; a higher level node may have no relationship with many of the lower level nodes.
>
> 3) With regards to your point about only using 1000 data points for a neural network: It was 1000 data points for the test set. We train the lstm for dataset sizes upto 12000, as mentioned in the Evaluation section, and in figures 4a and 5a. However, the simpler feedforward NN was trained with only 90 data points. We would like to highlight that the feedforward neural network only had 29 parameters to train (as shown in Figure 3), and was a simple single hidden layer network. We did train for multiple epochs to get convergence.
> Moreover, we wanted to learn with as few data points as possible to evaluate our active learning strategy, especially since we expect the data to come from a single user and thus be scarce. This is why we adopt feature-based active learning and the simple NN.
>
> 4) Regarding the test and validation sets: We did have a separate test set. We did not use a validation set for hyperparameter tuning. Given the scarcity of data in an active learning setting, we did not think to do so.
>
> 5) Resampling and Data Balancing: There are not separate classes in the data points, and so we did not think about resampling to balance the dataset. We did verify that there were plan traces across the range of preference values [0,1]. Perhaps resampling based on discrete sections in the range of values (eg: [0,0.2], [0.2,0.4], [0.4,0.6] etc)  might help, and we will try it.
>
> 6) Regarding your point on causality: Since we get the relevant features annotated by the user, we assume that those are the causal features for the user’s preferences. Then explanations about the preference value becomes about the effect of each feature on the preference/utility score. This is what we measure with the Attribution Error. The notion of interpretability that we considered is “...the degree to which a human can consistently predict the model’s result” as stated in the work of Been K. , Khanna R., and Koyejo R. (http://papers.nips.cc/paper/6300-examples-are-not-enough-learn-to-criticize-criticism-for-interpretability). By dropping or changing features, if the model’s prediction matches what the human would predict based on their own model of preferences, then it is more interpretable.

---

### Decision · Program_Chairs · 2019-05-15

**Decision:**

Accept

**Comment:**

While the reviewers do not fully agree on the decision, in the spirit of making the workshop a venue for discussion and feedback we decided to reject only those papers with strong reject votes.

Please address all review criticism as best possible for the final paper version and its presentation at the workshop. Looking forward to discuss your work at the workshop!